# International Multicentre Study of *Candida auris* Infections

**DOI:** 10.3390/jof7100878

**Published:** 2021-10-19

**Authors:** Nirav Pandya, Yasemin Cag, Nenad Pandak, Abdullah Umut Pekok, Aruna Poojary, Folusakin Ayoade, Teresa Fasciana, Anna Giammanco, Hulya Caskurlu, Dhanji P. Rajani, Yogesh Kumar Gupta, Ilker Inanc Balkan, Ejaz Ahmed Khan, Hakan Erdem

**Affiliations:** 1Consultant Microbiologist & Infection Preventionist, Bhailal Amin General Hospital, Vadodara 390003, India; drniravpandya@gmail.com; 2Department of Infectious Diseases & Clinical Microbiology, Istanbul Medeniyet University, Faculty of Medicine, Istanbul 34734, Turkey; dryasemincag@gmail.com (Y.C.); hcaskurlu@hotmail.com (H.C.); 3The Royal Hospital, Muscat 111, Oman; npandak@gmail.com; 4VM Medical Park, Pendik Hospital, Istanbul 34899, Turkey; umutpekok@yahoo.com.tr; 5Breach Candy Hospital Trust, Mumbai 400026, India; arunapoojary@gmail.com; 6Division of Infectious Diseases, University of Miami, Miller School of Medicine, Miami, FL 33136, USA; fxa375@med.miami.edu; 7Department of Health Promotion, Mother and Child Care, Internal Medicine and Medical Specialities, University of Palermo, 90133 Palermo, Italy; teresa.fasciana@virgilio.it (T.F.); anna.giammanco@unipa.it (A.G.); 8Microcare Laboratory & TRC, Surat 395003, India; microcaresurat@gmail.com; 9Fortis Escorts Hospital, Jaipur 302017, India; dr.yogeshmicro@gmail.com; 10Cerrahpaşa Medical School, Istanbul University, Cerrahpaşa 34096, Turkey; ilkerinancbalkan@hotmail.com; 11Shifa International Hospital, Shifa Tameer e Millat University, Islamabad 44000, Pakistan; ejazkhan99@hotmail.com; 12Department of Infectious Diseases, Bahrain Oncology Center, King Hamad University Hospital, Busaiteen 24343, Bahrain

**Keywords:** *Candida*, *C. auris*, fungi, outbreak, nosocomial, resistance

## Abstract

**Background:***Candida auris* has emerged globally as a multi-drug resistant yeast and is commonly associated with nosocomial outbreaks in ICUs. **Methods:** We conducted a retrospective observational multicentre study to determine the epidemiology of *C. auris* infections, its management strategies, patient outcomes, and infection prevention and control practices across 10 centres from five countries. **Results:** Significant risk factors for *C. auris* infection include the age group of 61–70 years (39%), recent history of ICU admission (63%), diabetes (63%), renal failure (52%), presence of CVC (91%) and previous history of antibiotic treatment (96%). *C. auris* was commonly isolated from blood (76%). Echinocandins were the most sensitive drugs. Most common antifungals used for treatment were caspofungin (40%), anidulafungin (28%) and micafungin (15%). The median duration of treatment was 20 days. Source removal was conductedin 74% patients. All-cause crude mortality rate after 30 days was 37%. Antifungal therapy was associated with a reduction in mortality (OR:0.27) and so was source removal (OR:0.74). Contact isolation precautions were followed in 87% patients. **Conclusions:**
*C. auris* infection carries a high risk for associated mortality. The organism is mainly resistant to most azoles and even amphotericin-B. Targeted antifungal therapy, mainly an echinocandin, and source control are the prominent therapeutic approaches.

## 1. Introduction

*Candida auris* is an emerging multidrug-resistant yeast that is spreading rapidly worldwide [1]. Genetic analyses show that five genetically different clades of *C. auris* emerged simultaneously from diverse geographical sites of different continents [2]. Infections caused by Candida species including *C. auris* are commonly detected in patients with prolonged hospitalization, especially in intensive care units (ICUs) [2,3]. It causes diseases ranging from superficial skin infections to invasive bloodstream infections (BSI) with high mortality rates (30% to 60%) [4]. In addition, the difficulty faced in the identification, incorrect use of antifungal drugs, and treatment failure are associated with a high mortality rate [5]. *C. auris* is associated with nosocomial outbreaks, in intensive care settings, and transmission—despite the implementation of enhanced infection prevention and control (IPC) measures—is a particular concern [6,7]. Some strains of the organism are resistant to the three major classes of antifungals, severely limiting treatment options [8]. Furthermore, the relationship between minimal inhibitory concentration (MIC) values and clinical outcomes is still not completely assumed, resulting in a lack of consensus on the susceptibility breakpoints for *C. auris* [9].

The aims of this multicentre study are to analyse the epidemiology of *C. auris* infections, to study prevalence of various potential risk factors, its diagnosis and management strategies, patient outcomes and infection prevention and control (IPC) practices in hospitals across countries.

## 2. Materials and Methods

This was a retrospective multicentre study through ID-IRI platform (Infectious Disease International Research Initiative). Ethical approval was obtained from the local ethics committee on 27 May 2019 (Approval code: 2019-39).

**Data Collection:** All patients with *C. auris* hospitalized in participant centres were included in the study and their data were submitted. An online questionnaire was prepared via Google Forms for each enrolled patient. In each centre, data were submitted by the researcher who is committed as per institutional agreement to collaborate in the study. The ten participating centres (Bhailal Amin General Hospital, Vadodara, India; The Royal Hospital, Muscat, Oman; Pendik Hospital, Bahcesehir University, Istanbul, Turkey; Breach Candy Hospital Trust Mumbai, India; University of Miami Hospital, Miami, Florida, USA; Surat Gastroenterology Hospital, Surat, India; Krishna Super specialty Hospital, Surat, India; Fortis Escorts Hospital Jaipur, India; Istanbul University-Cerrahpaşa, Cerrahpaşa Medical School, Turkey; Shifa International Hospital, Islamabad, Pakistan) submitted the data of *C. auris* isolates recovered from all clinical specimens collected between 1 January 2019 and 31 December 2019 through Google Forms. The form included parameters of demographics, risk factors, laboratory testing, treatment, outcome and infection control practices(Annexure-1).

Inclusion/Exclusion criteria:When multiple specimens of a patient showed growth of *C. auris*, only the specimen which was identified as source of infection was considered.When *C. auris* was isolated from multiple cultures of a single patient, only one isolate, preferably from a sterile body site, was included in the study.

**Identification:** In seven centres, identification of the fungus was performed with Vitek-2 according to the manufacturer’s instructions, using the latest software update (version 8.01), while in three centres, it was performed by matrix-assisted laser desorption ionization–time of flight mass spectrometry (MALDI-TOF) (Vitek mass spectrometer [MS], clinical knowledge database v3.2; bioMérieux).

**Determining resistance:** The antifungal susceptibility testing (AFST) was determined using two different methods: the broth microdilution method (BMD) (used by 2 centres) and Vitek 2 Compact system (used by 8 centres).

Clinical and Laboratory Standards Institute broth microdilution method (CLSI-BMD), following the M27-A3 guidelines. The antifungals tested were amphotericin B, fluconazole, voriconazole, caspofungin, micafungin, and anidulafungin.Vitek-2 Compact system using an AST-YS07 or AST-YS08 card, which tests the MIC of the six antifungals (Amphotericin B, Voriconazole, Fluconazole, Caspofungin, and micafungin) [10,11].

All the *C. auris* isolates were tested as per the manufacturer’s instructions. There are no established CLSI/EUCAST breakpoints for *C. auris*. However, the CDC has provided tentative breakpoints and guidance for *C. auris* MIC interpretation, based on information collected for *Candida* spp. and numerous expert opinions [12,13]. These tentative breakpoints were used for interpretation of sensitivity results for micafungin, anidulafungin, caspofungin, amphotericin B and fluconazole. The MIC distribution of *Candida auris* isolates from various centres was also studied.

Outcome analysis was performed as 30 days outcome and defined as:a.Resolution of infection and discharge of the patient;b.Infection not resolved but patient was discharged;c.Infection not resolved and patient was still hospitalized;d.Patient transferred to other hospital;e.Discharge against medical advice;f.Death of the patient.

Infection control practices such as the screening of patients and their close healthcare contacts for multisite colonization of *C. auris*, appropriate transmission-based precautions and environmental disinfection were performed and analysed as per CDC recommendations [11]. Statistical analysis with calculation of percentage and odds ratio (OR) was performed using Microsoft Office Excel 2007^®^ software (Redmond, WA, USA).

## 3. Results

A total of ten institutions from five countries participated in the study and submitted data for 54 patients. Data were analysed for risk factors, microbiology, treatment, patient outcomes and infection control practices. The most common age group was 61–70 (39%) years, followed by 71–80 (20%) years, 51–60 (17%) years and 81–91 (11%) years. Only 6% cases belonged to the <40 years of age group. The median age was 64.5 years. A total of 25 patients (46%) were females. A total of 37 patients (69%) were in the intensive care unit (ICU) and 17 were (32%) in the wards. A total of 40 patients (74%) had previous hospitalization history within the last 90 days, of which 34 patients (63%) had a recent history of ICU admission.

### 3.1. Risk Factors

Comorbid conditions: The most common risk factors were diabetes (*n* = 34, 63% patients), followed by renal failure (*n* = 28, 52% patients), chronic obstructive pulmonary disease (COPD) or chronic pulmonary illness (*n* = 21, 39% patients) and congestive heart failure (*n* = 16, 30%). A total of 14 patients (26%) had one or other type of immunocompromised condition such asunderlying malignancy, HIV, severe combined immunodeficiency (SCID), liver transplant, non-Hodgkin lymphoma (NHL), or systemic lupus erythematosus (SLE).

Invasive Devices/procedures: 91% (*n* = 49) of patients had a central venous catheter, 80% (*n* = 43) had a urinary catheter, 69% (*n* = 37) were on invasive ventilation, 37% (*n* = 20) were on haemodialysis, and 24% (*n* = 13) had a drain.

Therapeutic risk factors: The most common potential therapeutic risk factor for *C. auris* acquisition was a recent history of antibiotic treatment (*n* = 52, 96%). Other probable risk factors were corticosteroid therapy (*n* = 19, 35%), recent history of antifungal therapy (*n* = 15, 28%), total parenteral nutrition (*n* = 12, 22%), chemotherapy (*n* = 4, 7%), and immunomodulatory therapy (*n* = 2, 4%). Among 15 patients with a history of recent antifungal therapy, duration of antifungal exposure was more than 7 days in 73% (*n* = 11) of patients. The most common antifungals used were echinocandins (*n* = 8, 53%) and azoles (*n* = 6, 40%). Only one patient (7%) had a history of amphotericin-B therapy

ICU Stay: Out of 54 patients, 35 patients (65%) had a hospital stay of more than one month. A total of 23 patients (43%) had a stay of 1–2 months, while 12 patients (22%) had a stay of more than 3 months. In total, 39 out of 54 patients had an ICU stay (72%), while the remaining 15 patients did not have any ICU stay. Of these 39 ICU patients, 25 patients (90%) had an ICU stay of more than 1 week.

### 3.2. Microbiological Diagnosis

Analysis of culture and sensitivity report of 54 *C. auris* isolates was performed. *C. auris* was most commonly isolated from blood (*n* = 41, 76%). Other samples were skin and soft tissue (*n* = 6, 11%), respiratory tract (*n* = 5, 9%) and urine (*n* = 2, 4%). Overall susceptibility to antifungal drugs was as shown in Figure 1. In this study, echinocandins were the most sensitive drugs followed by amphotericin-B and azoles.

The minimum inhibitory concentration (MIC) data for *C. auris* isolates was provided by all the centres except one. MIC distribution of *C. auris* against eight antifungals is detailed in Table 1.

The CDC has not provided MIC breakpoints for voriconazole, posaconazole and 5-flucytosine against *C. auris* in its guideline document (12). Therefore, interpretation was not performed for these antifungals. However, as shown in Table 1, MIC distribution of these antifungals was determined. Notably, modal MIC of voriconazole was significantly lower (1 mg/L) than that of fluconazole (32 mg/L). Among echinocandins, modal MICs of micafungin, caspofungin and anidulafungin were 0.125, 0.25 and 1 mg/L, respectively. MIC50 and modal MICs for all the antifungals were the same except for anidulafungin. MIC50 and modal MICs for anidulafungin were 0.5 mg/L and 1 mg/L, respectively. MIC distribution of anidulafungin also showed an additional peak at MICs ≤0.06 mg/L.

The duration from admission to first positive culture for *C. auris* was also analysed (Table 2). The median time from admission to diagnosis (positive culture) was 20 days. A total of 26 patients (48%) also had concurrent growth of bacterial pathogens. Out of 26 patients, 12 patients (48%) had growth of bacterial pathogens in their blood, 6 (24%) in urine, 4 (16%) in respiratory tissue, and 3 (12%) in skin-soft tissue samples. Bacterial pathogens isolated from these patients are predominantly gram-negative organisms such as *Klebsiella pneumoniae*, *Escherichia coli* and *Pseudomonas aeruginosa*, as listed in Table 3. Repeat culture for *C. auris* was performed in 35 patients (65%). Out of 35, only 7 (20%) were subsequently positive for *C. auris*, while 28 (80%) turned out negative indicating microbiological clearance. Out of seven repeat positive patients, six were from blood and their repeat samples were taken within 7 days of first positive culture. In one patient repeat culture from post-CABG (coronary artery bypass grafting), chest wound was positive even after 63 days of the first positive culture, indicating wound colonization/chronic infection.

### 3.3. Therapeutic Issues

Out of total 54 patients, 47 were treated with antifungal therapy, while 7 patients did not receive any antifungals. Out of these seven, fourpatients expired or discharged before availability of reports and threepatients were managed only with source control viz. central venous catheter (CVC) removal in twopatients and abscess drainage in 1 patient.

Drug of choice: In the 47 patients treated with antifungals, drug of choice was echinocandins in 39 patients (83%) (caspofungin, *n* = 19, 40%; anidulafungin, *n* = 13, 28%; micafungin, *n* = 7, 15%). Five patients (11%) received fluconazole and one patient (2%) received voriconazole. Besides, one patient received liposomal amphotericin B, while the other received combination of caspofungin and liposomal amphotericin-B.

Treatment as per susceptibility testing: Out of 47 patients receiving antifungals, 30 patients (64%) were given treatment as per AFST report. Nine patients (19%) received antifungal (anidulafungin) as per sensitivity result of surrogate echinocandins (caspofungin). Eight patients (17%) received antifungals (seven echinocandins and one amphotericin-B) without AFST report, but based on identification of *C. auris*.

Dosing: 45 out of 47 patients received standard dosage of antifungals, as per CDC recommendations for *C. auris* [14]. Dose adjustment was conducted for Caspofungin in two patients with renal failure.

Duration of treatment: Out of 47 patients treated with antifungals, 25 patients (53%) received more than 14 days of antifungals. 10 (21%), 12 (26%), 17 (36%), 4 (9%) and 4 (9%) patients received <1 week, 1–2 weeks, 2–3 weeks, 3–4 weeks and >4 weeks of antifungal therapy, respectively. Out of 47 patients treated with antifungals, 13 patients (28%) died, 2 patients (4%) were discharged, and 2 patients (4%) were transferred to other hospital. On the whole, 24 patients expired, and the median duration of antifungal treatment in remaining 30 survivors was 20 days.

Reasons for stopping antifungals: Out of 47 patients who received antifungals, a total of 28 patients (60%) had microbiological clearance as evidenced by negative repeat culture. Reasons for stopping of antifungal treatment were microbiological clearance (51%, *n* = 24 patients), death (28%, *n* = 13 patients), improved patient condition (9%, *n* = 4), completion of treatment duration (4%, *n* = 2), discharge (4%, *n* = 2) and transferred to other hospital (4%, *n* = 2).

Source removal: Source removal was done in 40 out of 54 patients (74%). The most common approach was CVC removal within 24 h of positive culture in patients with Candidemia. Other approaches such as wound debridement and prosthetic material removal were performed, but mostly after 48 h of positive culture. Source removal was performed within 24 h in 60% patients in 24–48 h in 15% patients and after 48 h in 25% patients.

### 3.4. Outcome Analysis

The all-cause crude mortality rates at 30 days and at final outcome were 37% (*n* = 20) and 44% (*n* = 24), respectively. A total of 15 (28%) and 19 (35%) patients were discharged at 30 days and at the final outcome, respectively. Interestingly, the number of patients transferred to other hospitals or discharged against medical advice was 7% (*n* = 4) at 30 days which increased to 17% (*n* = 9) until final outcome.

Mortality: Out of 24 deaths, 20 patients (83%) died within 30 days of infection, while four deaths occurred after 30 days. Out of 24 deaths, 21 had bloodstream infections (88%), 2 had respiratory tract infections (8%) while 1had wound infection (4%). The 30-day crude mortality in *C. auris* candidemia was 44% (18 out of 41 patients). The mortality rate was higher (53%) in patients who had a hospital stay of <1 month (10 out of 19 patients) than (40%) in patients having a stay of >1 month (14 out of 35 patients). However, no similar effect of the duration of ICU stay was found on mortality rate. The mortality rate in patients with an ICU stay of <1 month was 50% (10 out of 20 patients), while it was 47% (9 out of 19 patients) in patients having an ICU stay of >1 month

Predictors of death: To find out any significant attributable risk factors for mortality in *C. auris* patients, risk factor comparison analysis was performed between two groups. Group 1—Expired patients vs. Group 2—Patients with other outcomes.

A. The prevalence of risk factors such as renal failure, congestive heart failure, haemodialysis, total parenteral nutrition and presence of devices such as invasive ventilator, central venous catheter was higher in Group1 (Expired patients) compared to Group2 (Patients with other outcomes) (See Table 4). However, diabetes did not have any significant impact on mortality (58% in Group-1 patients vs. 67% in Group-2 patients—OR: 0.70).

B. The mortality rate was higher in patients who either did not receive antifungal therapy (71%) or received <7 days of antifungals (65%), compared to patients who received antifungals >7 days (32%). Antifungal therapy was associated with a reduction inmortality (OR-0.27)

C. Source removal was associated with lower mortality (43% when source removal done vs. 50% when not done—OR: 0.74). Likewise, source removal was associated with a higher rate of resolution of infection (38% when source removal done vs. 21% when not done).

### 3.5. Infection Control Practice Analysis

Screening for multi-site colonization: Total 22 patients (41%) were screened. The most commonly screened sites were axilla (100%), groin (82%), nares (55%), oropharynx (46%), rectum (41%) and external ear canal (32%). Out of 22 patients, 18 (73%) were positive with colonization of at least one site, out of which, nine (41%) were colonized at two or more sites. The positivity rates for *C. auris* among various screening specimens were as depicted in Figure 2. Nares and groin samples showed highest sensitivity.

Screening of patient contacts: The contacts of 18 patients (33%) were screened for *C. auris* colonization. However, no contact was found to be positive.

Contact precautions for patients: Out of 54 patients, contact isolation precautions were followed in 47 patients (87%). Out of these 47 patients, 39 (83%) were placed in single isolation room, while 8 (17%) were kept in the cohort.

Discontinuation of contact precautions: See Table 5 for duration of contact isolation precautions among patients. Among 47 patients placed in contact precautions, in 35 patients (75%), contact precautions were followed until the final outcome (death—38%, discharge—36%). Other common reasons for discontinuation were microbiological clearance (17%) and source control with treatment completion (6%).

Surface disinfection: Out of 54 patients, in 49 patients (91%), chlorine-based solutions with available chlorine of >1000 ppm were used. Phenol (5%) was used in three patients (6%), while no specific disinfectants solutions could be used in two patients as the patients were either expired or discharged before the availability of the reports.

## 4. Discussion

The early detection of *C. auris* infections has been shown to be beneficial, as earlier initiation of appropriate antifungal therapy saved many lives [15]. Risk factors were not different from those associated with invasive infection due to other *Candida* spp. [6]. The most common associated risk factors in our study were diabetes mellitus, congestive cardiac failure, chronic kidney disease, chronic pulmonary illness, presence of CVC, urinary catheterization, post-operative drain, haemodialysis, invasive ventilation, a recent history of antibiotics and antifungal agents, chemotherapy, corticosteroid therapy, total parenteral nutrition and recent hospital/ICU stay.

Most of the reported cases of *C. auris* were isolated from blood. Other common clinical conditions include urinary tract infection, otitis, surgical wound infections, skin abscesses, peritonitis and wound infections [16,17]. In our study as well, *C. auris* was most commonly isolated from blood (76%), followed by skin and soft tissue (11%). The median time from admission to diagnosis (positive culture) was 20 days in our study, which is similar to 19 days in previous studies [18]. In our study, the order of resistance (azoles > amphotericin B > echinocandins) was the same as most of the studies [6] (Table 6). However, in the present study, resistance to amphotericin B and caspofungin is significantly higher than in other studies [19]. Due to the relatively low resistance to echinocandins, it is recommended that an echinocandin empirical therapy be initiated in patients suspected of having *C. auris* infections, particularly in patients with risk factors for *C. auris* candidemia [20]. However, the drug of choice will depend on the drug susceptibility report of the isolate. In our study, 83% of patients received echinocandins, while 13% of patients received azoles as per sensitivity report; meanwhile, in a study by Arensman et al., 92% of patients received echinocandins and 8% received azoles [21]. There is currently no evidence to support combination therapy in bloodstream infections with this organism, although if the urinary tract or CNS is involved dual therapy may be necessary [22]. In our study, only one patient with Candidemia received dual antifungals: that of caspofungin and liposomal amphotericin B.

The duration of antifungal treatment appears to be similar to those used for infections caused by other *Candida* spp. The duration depends on microbiological clearance, clinical cureand source removal. In our study, the median duration for microbiological clearance was 20 days. Source removal approaches such as CVC removal, wound debridement has a favourable impact on reducing the antifungal duration and also in improving clinical outcomes. Thus, source removal should be done as early as possible after diagnosis.

The all-cause 30-day crude mortality rate was 37% in our study, which is similar to 35.2% reported by Morales-López et al. [25]. In the present study, 30-day crude mortality in *C. auris* candidemia was 44%, which is similar to 41.9% of the study by Rudramurthy et al. [17], but lower than 61.1% of the study of Shashtri et al. [26]. Candidemia was associated with higher mortality (OR: 3.5), similar to the study of Sayeed et al. (OR-4.3). Bacterial co-infection was also associated with higher mortality (OR-2.1), similar to the study by Sayeed et al. (OR: 2.3) [6,16,17,18,19,20,21,22,25]. The risk factors that are significantly associated with higher mortality were congestive heart failure, renal failure, haemodialysis, invasive ventilator, CVCand total parenteral nutrition. However, diabetes was not associated with higher mortality (OR: 0.7), which is in contrast to other studies such as Sayeed et al. (OR-2.3) [16]. Considering management strategies, antifungal therapy and source removal were both associated with a significant reduction in mortality in our study. This is in line with clinical practice guidelines for treating invasive candidiasis [21].

For screening of patients for multi-site colonization of candida, CDC recommends axilla, groinand sometimes nares, while Public Health England recommends groin, axilla, urine, nose, throat, perineal swab, rectal swabor stool sample [5]. In our study, as the groin, nares and axilla have the highest positivity rate for colonization, we recommend including at least these three sites for screening. Close healthcare contacts of patients with newly identified *C. auris* infection or colonization should be considered for screening for *C. auris* colonization [27]. Identifying persons colonized with *C. auris* is a key step in containing the spread of *C. auris*. In addition, this is a useful tool for outbreak investigations. Patients on contact precautions for *C. auris* should be placed either in a single room or in cohort with patients with *C. auris* [28]. The CDC recommends continuing proper transmission-based precautions for the entire duration of the patient’s stay in the facility. For terminal cleaning of a bedspace or room vacated by a *C. auris* patient, currently, hypochlorite solution is recommended at 1000 ppm of available chlorine [29]. In a comparison of the efficacies of a range of disinfectants, sodium hypochlorite and hydrogen peroxide resulted in the greatest reduction in *C. auris* CFU (colony forming units) [30].

In conclusion, the present study confirms that *C. auris* is an emerging multidrug-resistant yeast that presents a great challenge to global health, especially in the hospital environment. Our study has certain limitations that are inherent to a retrospective, observational study. The sample size to analyse the antifungal MIC distribution of *C. auris* isolates is not large. Furthermore, sample sizes from various centres were not big enough to allow the comparison of study objectives between centres. The study is not able to provide an attributable mortality rate due to *C. auris* infection. However, the present study confirms that *C. auris* infection carries a high risk ofmortality with candidemia in particular. *C. auris* isolates are largely resistant to most azoles and even to amphotericin B. Echinocandins are the most sensitive antifungals and, thus, are the first drugs of choice. Targeted anti-fungal therapy, primarily echinocandin and source control, are the most important therapeutic approaches to reduce mortality. Infection control practices such as the screening of close contacts, contact precautions and terminal cleaning are of utmost importance to identify and control the spread of *C. auris* in the hospital setup.

## Figures and Tables

**Figure 1 jof-07-00878-f001:**
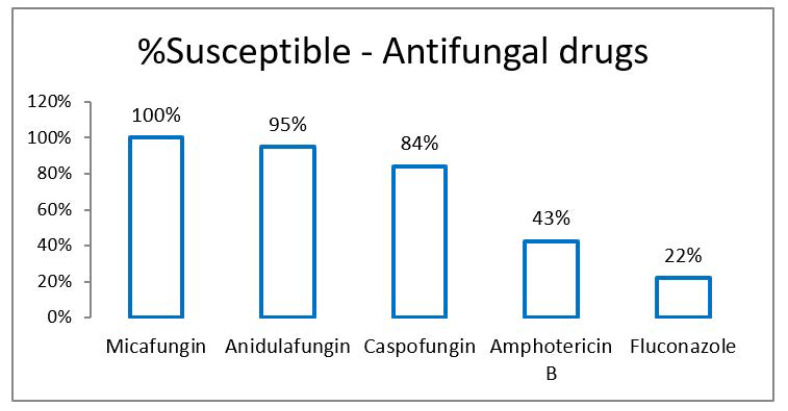
Susceptibility (%) of antifungal drugs in *Candida auris*.

**Figure 2 jof-07-00878-f002:**
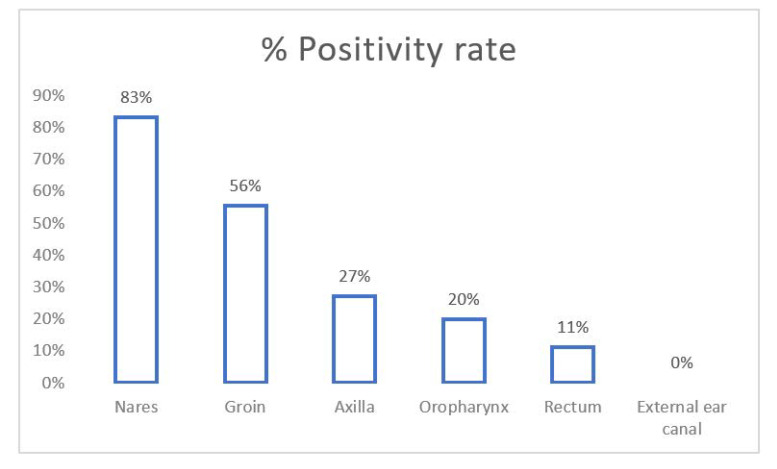
Positivity rate (%) of various samples for screening of *Candida auris* colonization in patients.

**Table 1 jof-07-00878-t001:** MIC distribution of *C. auris* isolates.

	≤0.06	0.12	0.25	0.5	1	2	4	8	16	32	64	128	≥256	Total (*n*)	MIC Range	GM ^a^	MIC50 ^b^	MIC90 ^c^
Fluconazole								8	4	11 ^d^	1		6	30	8–≥256	4.62	32	≥256
Voriconazole		2	1	1	12	2	7	5						30	≤0.12–≥8	2.89	1	≥8
Amphotericin B				6	10	4	2	1	7					30	0.5–≥16	3.87	1	≥16
Caspofungin		2	18	2	4	3	4							33	0.12–4	3.89	0.25	4
Micafungin	6	14		2	6	4								32	≤0.06–2	5.26	0.12	2
Anidulafungin	4	2	2	2	4	3	2							19	≤0.06–4	2.58	0.5	4
Posaconazole	3		1				9							13	0.06–4	3.00	4	4
5-Flucytosine					3				4	1	8			16	≤1–≥64	3.13	32	≥64

^a^ GM, geometric mean MICs. ^b^ MIC50, MIC at which 50% of test isolates were inhibited. ^c^ MIC90, MIC at which 90% of test isolates were inhibited. ^d^ Modal MICs (most repeated MIC value) are indicated with underlined numbers.

**Table 2 jof-07-00878-t002:** Time from admission to positive culture.

No. of Days	Patient No.	Patient %
≤2 days	5	9%
3–7 days	8	15%
8–14 days	8	15%
15–30 days	17	31%
>1 month	16	30%

**Table 3 jof-07-00878-t003:** Bacterial pathogens isolated from *C. auris* patients.

Organism	No. of Organism	% of Organisms
*Klebsiella pneumoniae*	7	24%
*Escherichia coli*	6	21%
*Pseudomonas aeruginosa*	5	17%
*Staphylococcus aureus*	3	10%
*Acinetobacter baumanii*	2	7%
*Burkholderia cepacia*	2	7%
*Stenotrophomonas maltophilia*	1	3%
*Acinetobacter calcoaceticus*	1	3%
*Enterococcus faecalis*	1	3%
*Klebsiella ozaenae*	1	3%
Total	29	

**Table 4 jof-07-00878-t004:** Analysis to determine the risk factors for mortality among *C. auris* cases.

Risk Factor	Group-1 (Expired Patients)	Group-2 (Patients with Other Outcome)	Odds Ratio
Renal failure	67%	40%	3.0
Congestive Heart Failure	46%	17%	4.23
Invasive ventilator	75%	63%	1.74
Haemodialysis	63%	17%	8.33
Total parenteral Nutrition	33%	13%	3.25
Central Venous Catheter	100%	83%	4.60
Candidemia	88%	67%	3.5
Bacterial co-infection	58%	40%	2.1

**Table 5 jof-07-00878-t005:** Duration of contact isolation precautions.

Days	Total	%
≤1 week	15	32%
1 week–1 month	23	49%
1–2 month	3	6%
2–3 months	1	2%
>3 months	5	11%
Grand Total	47	100%

**Table 6 jof-07-00878-t006:** Comparison table of resistance patterns for *C. auris* (% of Resistant).

Antifungal	Present Study	Osei et al. [19]	Lockhart et al. [23]	Chowdhary et al. [20]	Chen et al. [24]	CDC Report [11]
Micafungin	0%	1.25%	7%	2%	0.8%	<5%
Anidulafungin	5%	1.25%	7%	2%	1.1%	<5%
Caspofungin	16%	3.48%	7%	2%	12.1%	<5%
Amphotericin B	58%	15.46%	35%	8%	12.%	30%
Fluconazole	78%	44.29%	93%	90%	91%	90%

## Data Availability

Not applicable.

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
