# Peer review of "International Multicentre Study of Candida auris Infections"

_jof, 2021, doi:10.3390/jof7100878_

Round 1

Reviewer 1 Report

The authors reported on an international multicentre study of Candida auris infections with the aim to identify risk factors. Candida auris is an emerging pathogen and its control becomes increasingly important. Because of this a number of clinical studies have already been published addressing several aspects of this pathogen but many of these studies have not been discussed by the authors. E. g. Shastri, P. S., Shankarnarayan, S. A., Oberoi, J., Rudramurthy, S. M., Wattal, C., & Chakrabarti, A. (2020). Candida auris candidaemia in an intensive care unit–Prospective observational study to evaluate epidemiology, risk factors, and outcome. Journal of critical care, 57, 42-48; Chen, J., Tian, S., Han, X., Chu, Y., Wang, Q., Zhou, B., & Shang, H. (2020). Is the superbug fungus really so scary? A systematic review and meta-analysis of global epidemiology and mortality of Candida auris. BMC infectious diseases, 20(1), 1-10; Tian, S., Bing, J., Chu, Y., Chen, J., Cheng, S., Wang, Q., ... & Shang, H. (2021). Genomic epidemiology of Candida auris in a general hospital in Shenyang, China: a three-year surveillance study. Emerging microbes & infections, accepted, 1-28; and others. How do their findings fit into the results of the presented study?

The risk factors given are a mixture of risk factors and treatment preferences. There is no proof that e.g. hydrocortisone is a risk factor for C. auris infections. The authors have to define first what a risk factor is, how it will be identified and only then they can analyze their data applying these criteria. Here they have to be aware that 54 patients are a rather low number and the statistical significance will soon run thin.

When using computer programs one get really nice calculations but the results has to be set into the context at hand. For the entire manuscript and e. g. Graph 1, tables, etc.: please be more realistic: with only 54 patients you cannot determine susceptibility down to two decimals! One patient more or less would change the percentage by almost 2%, therefore, any decimals are meaningless here. This is even more critical for table 3 with only 29 isolates. Just percentages without any decimals would be fine throughout the entire manuscript.

Lines 120ff: place put a space, e. g. between n=34,62.96% to n=34, 62.96% and so on

Paragraph starting with l. 324: why is C. auris here in bold?

Author Response

Comment

The authors reported on an international multicentre study of Candida auris infections with the aim to identify risk factors. Candida auris is an emerging pathogen and its control becomes increasingly important. Because of this a number of clinical studies have already been published addressing several aspects of this pathogen but many of these studies have not been discussed by the authors. E. g.

1.      Shastri, P. S., Shankarnarayan, S. A., Oberoi, J., Rudramurthy, S. M., Wattal, C., & Chakrabarti, A. (2020). Candida auris candidemia in an intensive care unit–Prospective observational study to evaluate epidemiology, risk factors, and outcome. Journal of critical care57, 42-48;

2.      Chen, J., Tian, S., Han, X., Chu, Y., Wang, Q., Zhou, B., & Shang, H. (2020). Is the superbug fungus really so scary? A systematic review and meta-analysis of global epidemiology and mortality of Candida auris. BMC infectious diseases20(1), 1-10;

3.      Tian, S., Bing, J., Chu, Y., Chen, J., Cheng, S., Wang, Q., ... & Shang, H. (2021). Genomic epidemiology of Candida auris in a general hospital in Shenyang, China: a three-year surveillance study. Emerging microbes & infections, accepted, 1-28; and others.

How do their findings fit into the results of the presented study?

Response

Thanks for the valuable references. All the mentioned & few other reference articles were reviewed. We incorporated important & significant findings of these studies into present study article e.g. 30-day crude mortality rate of 61.1% in C. auris candidemia from the study of Shastri et al in discussion section & antifungal resistance profile of C.auris isolates from the study of Chen et al. in Table 6.

Comment

The risk factors given are a mixture of risk factors and treatment preferences. There is no proof that e.g. hydrocortisone is a risk factor for C. auris infections. The authors have to define first what a risk factor is, how it will be identified and only then they can analyze their data applying these criteria. Here they have to be aware that 54 patients are a rather low number and the statistical significance will soon run thin.

Response

Aim of our study is not to establish causal association between risk factor & C. auris infections but to observe & study prevalence of various potential risk factors among C. auris patients. (We have added this in Introduction section). We tried to study all the possible risk factors for Candida infections as described in published literature.

Though corticosteroid therapy is a risk factor for invasive fungal infections such as Candidiasis, hydrocortisone cannot be proven as a more significant risk factor than other glucocorticoids. Also as pointed out, our study has limitations in sample size to establish this. So we have removed the comparison of hydorcortisone&dexamethsaone as risk factors from RESULTS section.

Comment

When using computer programs one get really nice calculations but the results has to be set into the context at hand. For the entire manuscript and e. g. Graph 1, tables, etc.: please be more realistic: with only 54 patients you cannot determine susceptibility down to two decimals! One patient more or less would change the percentage by almost 2%, therefore, any decimals are meaningless here. This is even more critical for table 3 with only 29 isolates. Just percentages without any decimals would be fine throughout the entire manuscript.

Response

Corrected. Thanks for the valuable suggestion.

Comment

Lines 120ff: place put a space, e. g. between n=34,62.96% to n=34, 62.96% and so on

Response

Corrected. Thanks for the valuable suggestion.

Comment

Paragraph starting with l. 324: why is C. auris here in bold?

Response

Corrected. Thanks for the valuable suggestion.

Reviewer 2 Report

This is a comprehensive retrospective study on C. auris cases in multiple locations. The findings are not surprising but supportive of most of what is known about C. auris with respect to antifungal resistance and risk factors. Since there is still little known about this emerging fungus, this adds to the story. The limitation is the small number of the cases and thus the transference to a general picture of C. auris. 

A few minor comments.

It would be interesting to note if there were any higher correlations of antifungal susceptibility patterns, time of acquisition, time of clearance, with geographical location of the hospital/ patients.

In materials andn methods - line 73 - not sure what this means. Do you mean that only one isolate was included in the study? Also it is not clear what criteria 3 is. 

Table I could be improved by adding column lines.

Author Response

Comment

It would be interesting to note if there were any higher correlations of antifungal susceptibility patterns, time of acquisition, time of clearance, with geographical location of the hospital/ patients.

Response

The study analysed all these parameters for the study isolates. But sample sizes from various centres were not big enough to allow comparison of study objectives between geographic locations or centres. The same has been mentioned in study limitation.

Comment

In materials and methods - line 73 - not sure what this means. Do you mean that only one isolate was included in the study?

Response

Yes. Only one isolate per patient, preferable from sterile body site, was included in the study.

The sentence is modified in the article to elaborate this.

Comment

Also it is not clear what criteria 3 is. 

Response

Many times, patient having Candia auris infection, are screened for multi-site Candida colonization as a part of infection prevention & control (IPC) protocol. In such cases, Candida auris isolated from the screening culture samples like nares, axilla, groin etc. were excluded from the study because these isolates represent colonization only & not true infection.

The criterion 3 is removed from the article as it is covered into criteria 2 now.

Comment

Table I could be improved by adding column lines.

Response

Corrected. Thanks for the valuable suggestion.

Round 2

Reviewer 1 Report

Most of the suggestions have been addressed by the authors. However, one small item still remains. As I wrote before: Lines 14-170: place put a space, e. g. between (n=34,63% patients) to (n=34, 63% patients) and so on.

Author Response

Comment

Most of the suggestions have been addressed by the authors. However, one small item still remains. As I wrote before: Lines 14-170: place put a space, e. g. between (n=34,63% patients) to (n=34, 63% patients) and so on.

Response

Corrected. Previously it was missed at some places. Thanks for the valuable suggestion.